

# A new pseudoscorpion genus (Garypinoidea: Garypinidae) from the Eocene supports extinction and range contraction in the European paleobiota

Nova Stanczak[1,2], Mark S. Harvey[3,4], Danilo Harms[2,5,6], Jörg U. Hammel[7], Ulrich Kotthoff[8] and Stephanie F. Loria[2]

[1] Department of Biology, University of Hamburg, Hamburg, Germany
[2] Section Arachnology, Centre for Taxonomy and Morphology, Museum of Nature Hamburg—Zoology, Leibniz Institute for the Analysis of Biodiversity Change, Hamburg, Germany
[3] Collections & Research, Western Australian Museum, Welshpool, Australia
[4] University of Western Australia, Crawley, Australia
[5] Harry Butler Institute, Murdoch University, Murdoch, Australia
[6] Centre for Invasion Biology, University of Venda, Thohoyandou, South Africa
[7] Institute of Materials Physics, Helmholtz-Zentrum Hereon, Geesthacht, Germany
[8] Centre for Biomonitoring and Conservation Science, Museum of Nature Hamburg—Geology-Paleontology, Leibniz Institute for the Analysis of Biodiversity Change, Hamburg, Germany

Corresponding author
Stephanie F. Loria,
s.loria@leibniz-lib.de

## ABSTRACT

During the Paleogene, the Holarctic experienced drastic climatic oscillations, including periods of extensive glaciation. These changes had a severe impact on both the flora and fauna causing widespread extinction and range shifts with some taxa retreating to refugia in the Mediterranean Basin. Here we provide evidence for this hypothesis using fossils from the pseudoscorpion family Garypinidae Daday, 1889 (Arachnida: Pseudoscorpiones). This family comprises 21 extant genera from all continents except Antarctica but is restricted to low mid-latitudes (<44°N) in the Northern Hemisphere. We provide the second record of garypinids from the European succinite ambers of the Eocene by describing the first extinct genus in Garypinidae, *Baltamblyolpium* gen. nov., which includes two species: *Baltamblyolpium gizmotum* sp. nov. from Baltic amber and *Baltamblyolpium grabenhorsti* sp. nov. from Bitterfeld amber. The new genus exhibits a morphology that closely resembles *Neoamblyolpium* Hoff, 1956 from western North America and the genus *Amblyolpium* Simon, 1898, which is widespread but includes taxa restricted to Mediterranean refugia in Europe. The discovery of a new fossil genus of Garypinidae from Europe confirms that the family was found at more northerly latitudes during the Eocene, however, extinction and range contraction resulted in their present-day relictual distribution in southern Europe like many other lineages that once thrived in the European "Baltic amber forest" of the Eocene.

## INTRODUCTION

The biota of the Holarctic has been shaped by drastic climatic shifts and fluctuations during the Cenozoic, most notably the severe glaciations of the Plio-Pleistocene that followed the warm-temperate Eocene and Miocene periods and affected all landscapes (*Schmitt, 2007*; *Kirschner et al., 2022*). These severe changes had significant and long-lasting effects on the fauna and flora of this region, causing range shifts that involved complex scenarios of dispersal, vicariance and extinction (*Enghoff, 1995*; *Hewitt, 1999*; *Sanmartín, Enghoff & Ronquist, 2001*; *Schmitt & Varga, 2012*). Across Central Europe, many taxa became extinct or retreated to southern latitudes. The 'Mediterranean glacial refugium' hypothesis (*Brito, 2005*) posits that the Mediterranean Basin, a biodiversity hotspot that includes the Iberian, Italian and Balkan Peninsulas and northern Africa, served as a refugium for many Central European lineages during periods of glaciation (*Provan & Bennett, 2008*; *Gentili et al., 2015*; *Di Pasquale et al., 2020*; *Krause et al., 2022*). The European succinite ambers (Baltic, Bitterfeld and Rovno) from the Eocene offer an exciting possibility to evaluate this hypothesis in a qualitative manner since they are, without a doubt, the richest deposits from the Cenozoic of Europe and contain diverse assemblages of invertebrates and plants no longer present in northern or Central Europe today (Fig. 1; *Sadowski et al., 2017*; *Bogri, Solodovnikov & Żyła, 2018*; *Dunlop et al., 2018*; *Penney, 2020*). Recent analyses of plant assemblages in Baltic amber have painted the picture of a warm temperate "Baltic amber forest" across the Eocene of Europe that was botanically diverse, had regional differences at the species level, and thrived under a relatively high precipitation regime (*Sadowski et al., 2017*, *2019*). Botanically, this forest was unlike the forests of Europe today but rather warm-temperate, thereby providing a context in which to evaluate biogeographical hypotheses for invertebrate taxa preserved in this amber type (*e.g.*, *Brunke et al., 2019*; *Schwarze et al., 2021*).

Pseudoscorpions (Arachnida: Pseudoscorpiones) are an ancient arachnid lineage with the oldest fossil, *Dracochela deprehendor* *Schawaller, Shear & Bonamo, 1991*, dating back to the Middle Devonian, 392 Ma (Fig. 2; *Shear, Schawaller & Bonamo, 1989*; *Schawaller, Shear & Bonamo, 1991*; *Judson, 2012*). Recent phylogenomic analyses (*Benavides et al., 2019*) and studies on Cretaceous amber (*Harvey et al., 2018*; *Porta et al., 2020*; *Wriedt et al., 2021*; *Geißler et al., 2022*) and Upper Triassic compression (*Kolesnikov et al., 2022*) fossils suggest that diversification of major lineages and families occurred in the Paleozoic. This implies that extinct species from the European succinite ambers of the Eocene mostly belong to extant genera and/or lineages (*Harms & Dunlop, 2017*). Pseudoscorpions from these ambers have been studied for more than a century (*Koch & Berendt, 1854*; *Menge, 1854*; *Beier, 1947*, *1955a*). Presently, 36 pseudoscorpion species in seven extinct and 11 extant genera belonging to 13 (52%) of the 25 recognized families have been described from all three European succinite amber types (*Harms & Dunlop, 2017*; *Schwarze et al., 2021*; *WPC, 2023*), which includes Baltic amber from the Gulf of Gdańsk, Bitterfeld amber from eastern Germany and Rovno amber from Ukraine (Fig. 1; *Wolfe et al., 2016*).
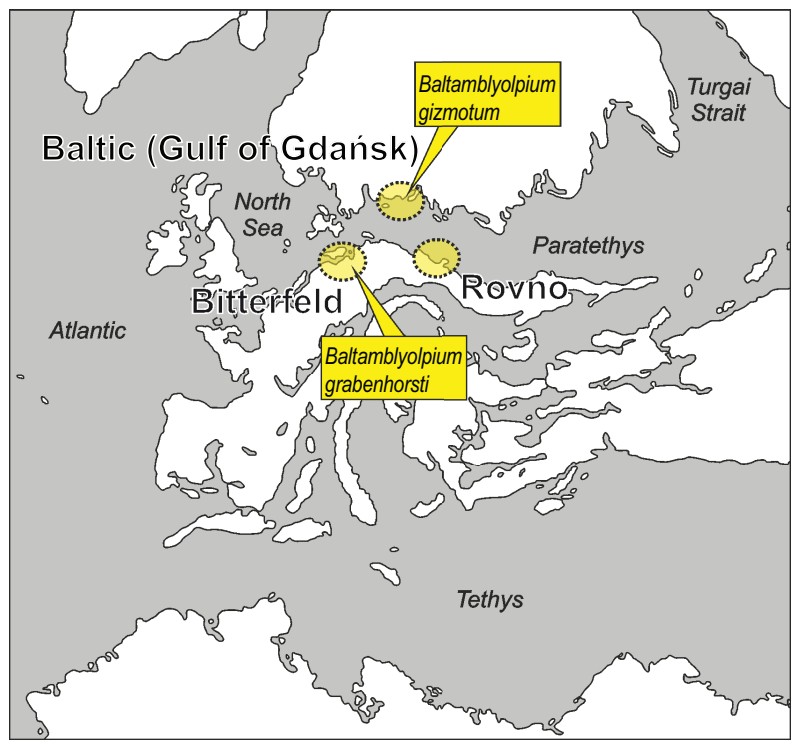

**Figure 1 Putative paleogeographical positions of source areas for Baltic, Bitterfeld and Rovno amber deposits in Europe from the early to middle Eocene.** Possible paleotype localities of *Baltamblyolpium gizmotum* sp. nov. and *Baltamblyolpium grabenhorsti* sp. nov. indicated. Modified after *Popov et al. (2004)*, *Denk & Grimm (2009)*, *Deep Time Maps (2020)*, *Szwedo & Sontag (2013)*, *Wolfe et al. (2016)* and *Dunlop et al. (2018)*.

The biogeographical picture for the European pseudoscorpion paleofauna is mixed. The genera of some families (*e.g.*, Chthoniidae *Daday, 1889* and Neobisiidae *Chamberlin, 1930*) preserved in Baltic amber are still widespread in Europe today (*Judson, 2003*), whereas others have gone extinct but have extant relatives across Europe (*e.g.*, Atemnidae Kishida, 1929, Cheliferidae Risso, 1827, Chernetidae Menge, 1855, Chthoniidae and Withiidae Chamberlin, 1931). Several pseudoscorpion families show clear indications of biogeographical shifts or range contraction since the Eocene. Feaellidae Ellingsen, 1906 occurs in Baltic amber and Triassic deposits in Ukraine but is now confined to former Gondwanan landmasses (*Harvey & Šťáhlavský, 2010*; *Harms & Dunlop, 2017*; *Novák, Lorenz & Harms, 2020*; *Schwarze et al., 2021*; *Kolesnikov et al., 2022*), while Geogarypidae *Chamberlin, 1930* is preserved in both Baltic and Rovno ambers but is now restricted to lower latitudes in southern Europe. Pseudotyrannochthoniidae Beier, 1932, and Pseudogarypidae Chamberlin, 1929, which are both found in Baltic amber, exhibit present-day disjunct distributions as they are absent in Europe today but have Recent relatives in North America and/or eastern Asia.

During an ongoing inventory of fossil pseudoscorpions, we recently discovered two fossils of the family Garypinidae *Daday, 1889* in both Baltic and Bitterfeld ambers.
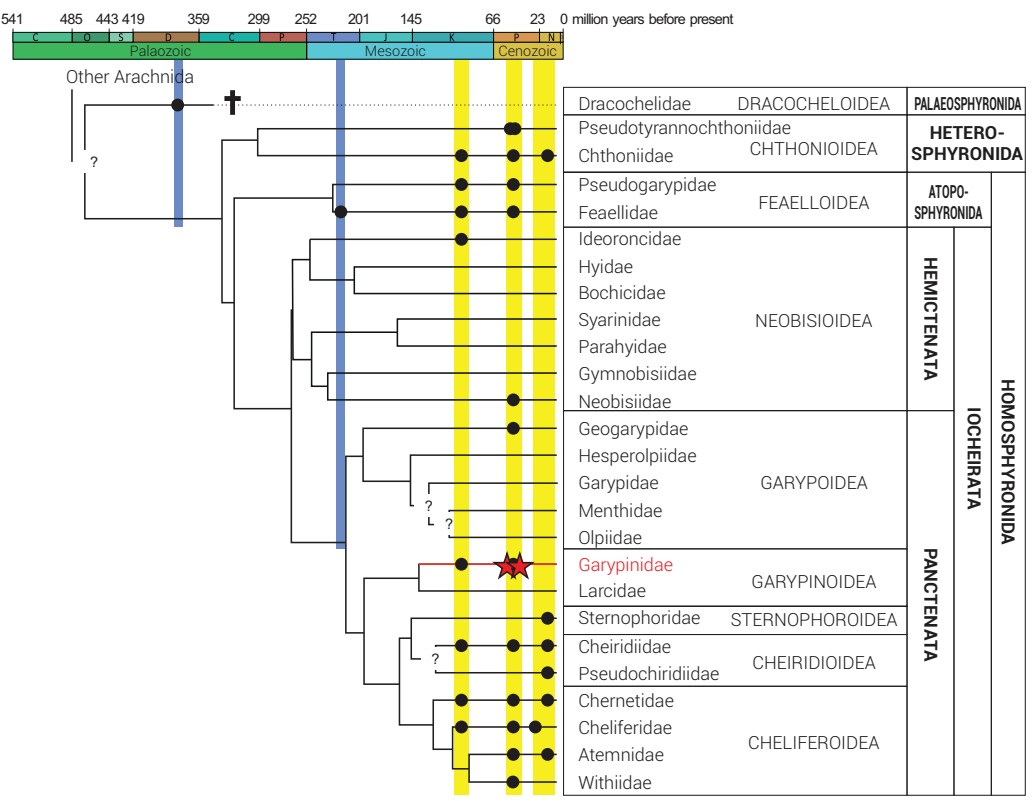

**Figure 2 Phylogeny of Pseudoscorpiones De Geer, 1778, following *Benavides et al. (2019)*.** Placement of previously described fossil species (black circles) and the two new Garypinidae *Daday, 1889* species (red stars), *Baltamblyolpium gizmotum* sp. nov. and *Baltamblyolpium grabenhorsti* sp. nov., indicated. Figure adapted from *Schwarze et al. (2021)*, *Geißler et al. (2022)* and *Johnson et al. (2023)*.

Garypinidae comprises 81 Recent (Holocene) and two fossil species in 21 Recent genera found across the globe but all extant species found in the Northern Hemisphere are restricted to low mid-latitudes (<44°N) (Fig. 3). Garypinidae includes two fossil species in extant genera: *Amblyolpium burmiticum* (*Cockerell, 1920*) from Upper Cretaceous Burmese amber, and *Garypinus electri Beier, 1937* from Eocene Baltic amber. Interestingly, both the Baltic and Bitterfeld amber fossils described here do not resemble Baltic *G. electri*, but rather are most similar to *Neoamblyolpium Hoff, 1956* from western North America and the widespread genus *Amblyolpium Simon, 1898*, which is found in southern Europe, the Middle East, South and Southeast Asia, Melanesia, Africa, the Caribbean and South America. In the present contribution, we describe the Baltic and Bitterfeld amber fossils as two new species and name a new genus, *Baltamblyolpium*, gen. nov. to accommodate them, providing detailed descriptions, photos, illustrations and 3D-reconstruction models based on synchrotron data. We suggest that the present-day, southerly distribution of Garypinidae in Europe can be explained by late Cenozoic climate change that caused extinction or range contraction into Mediterranean refugia of Central European lineages. This hypothesis corresponds with biogeographical studies across many other European taxa.

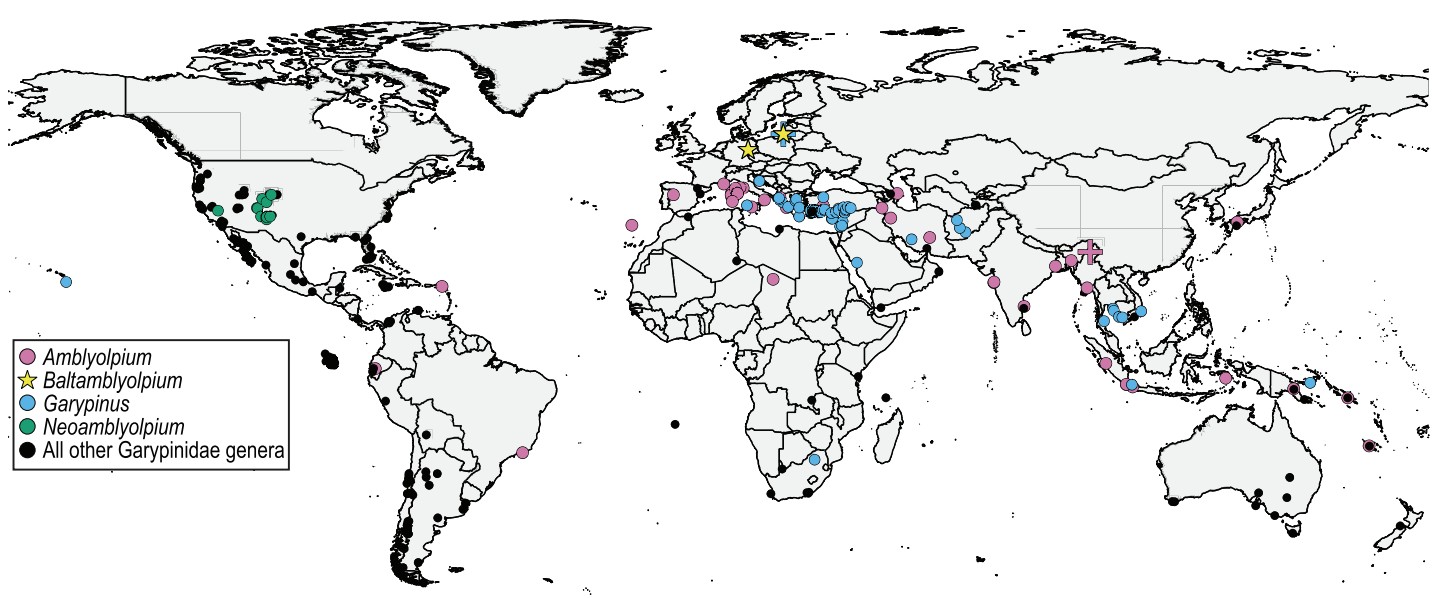

**Figure 3 Distribution of Garypinidae** *Daday, 1889*, **highlighting relevant genera (color) with previously described extant (circles) and fossil (crosses) species, and the two new fossil species (stars),** *Baltamblyolpium gizmotum* **sp. nov. and** *Baltamblyolpium grabenhorsti* **sp. nov.**

## MATERIALS AND METHODS

Two garypinid specimens (one from Baltic amber and one from Bitterfeld amber) were examined for the present study. The specimen in Baltic amber was purchased by Carsten Gröhn and placed in his personal collection, Carsten Gröhn Private Collection (CGPC), Glinde, Germany, before being deposited in the Geological and Paleontological Institute and Museum (GPIH), Museum of Nature, Leibniz Institute for the Analysis of Biodiversity Change, Hamburg, Germany in 2021–2022. The specimen in Bitterfeld amber was originally deposited in the Heinrich Grabenhorst Private Collection (HGPC), Wienhausen, Germany but is now deposited in the amber collection of the Geoscience Museum, Geoscience Center, University of Göttingen (GZG.BST), Germany.

To remove excess amber, the Baltic amber piece was wet-sanded with waterproof silicon carbid sandpaper of different grain sizes (Waterproof SiC, FEPA P #120, #1200, #4000, Struers GmbH). Because the Bitterfeld amber specimen is poorly preserved and broken into two heavily oxidized pieces that contain multiple cracks, it was not sanded. All images were taken using either a Leica M205 stereomicroscope with Leica Application Suite X software (v.3.0.1.15878), or the microptic stacking BK PLUS Lab Imaging System, Dun Inc. equipped with a Canon EOS 7D camera with Capture One Pro 9.3 64 Bit software (v.9.3.0.85). Depending on specimen size, microscopic (5× and 10× magnification) or macroscopic (65 mm) lenses were used for photography. During imaging, the Baltic amber piece was submerged in baby oil (Bübchen Baby Öl; Bübchen Werk Ewald Hermes, Pharmazeutische Fabrik GmbH, Soest, Germany), whereas the Bitterfeld amber piece was not due to its fragility. Images were stacked using Zerene Stacker (v.1.04; Zerene Systems

LLC, Richland, WA, USA) and edited in Adobe Photoshop CS6. Synchrotron radiated micro-computed tomography (SRµCT) scans were acquired for both pieces at the German Electron Synchrotron (Deutsches Elektronen-Synrchrotron, Hamburg, Germany) at PETRA III beamline P05 operated by the Helmholtz-Zentrum Hereon (*Greving et al., 2014*; *Wilde et al., 2016*). Images were taken with a photon energy of 17,999 eV, a sample distance to the detector of 0.079999 m, and an effective voxel size of 0.607836 µm in the reconstructed tomographic volume. Tomographic reconstruction of two times binned raw projections was done by applying a transport of intensity phase retrieval approach and using the Filtered Back Projection (FBP) algorithm implemented in a custom reconstruction pipeline (*Moosmann et al., 2014*) using Matlab (Math-Works) and the Astra Toolbox (*Palenstijn, Batenburg & Sijbers, 2011*; *van Aarle et al., 2015*, *2016*). Scan data was then converted into a 3D-model using SideFX Houdini 18.5.672 and 19.0.382 and Amira 6.0.2. Redshift 3.0.55 and Paraview 5.6.0 were utilized to render 3D-models and deposited in MorphoSource (https://www.morphosource.org; Project ID: 000490376). Digital drawings were made with the software Procreate 5.2.5 and drawn to scale. A distribution map was generated using occurrence data from the World Arachnid Catalog (*WPC, 2023*; *Harms et al., 2022*) with QGIS ver. 3.14 (http://www.qgis.org) by superimposing locality records onto political borders from Natural Earth (https://www.naturalearthdata.com). Terminology for morphological structures follows *Geißler et al. (2022)*. Obstructed structures are displayed with question marks (?). Measurements (in mm) were taken with a Leica M205 stereomicroscope using Leica Application Suite X. Measurements for the podomeres include width for the pedipalps (dorsal view) and depth for the legs (lateral view).

For details on the paleontological setting and description of amber types, please see *Sadowski et al. (2017)* or *Schwarze et al. (2021)*. Baltic amber is generally considered to be of Upper Eocene age (Lutetian: ca. 49–44 Ma; *Weitschat, Wichard & Penney, 2010*) although recent studies (*Sadowski et al., 2017*) have argued for a younger age (Bartonian–Rupelian: ca. 38–33.9 Ma). The age of Bitterfeld amber is more controversial (*Dunlop et al., 2018*; *Ahrens et al., 2019*), however, *Mänd et al. (2018)* suggested a roughly contemporaneous origin of all succinites based on stable isotope analyses. Please see *Schwarze et al. (2021)* for a full discussion on this matter.

The electronic version of this article in Portable Document Format (PDF) will represent a published work according to the International Commission on Zoological Nomenclature (ICZN), and hence the new names contained in the electronic version are effectively published under that Code from the electronic edition alone. This published work and the nomenclatural acts it contains have been registered in ZooBank, the online registration system for the ICZN. The ZooBank LSIDs (Life Science Identifiers) can be resolved and the associated information viewed through any standard web browser by appending the LSID to the prefix http://zoobank.org/. The LSID for this publication is: urn:lsid:zoobank.org:pub:8EC926A6-3F0B-46A9-BD5A-CF79CB4326B8. The online version of this work is archived and available from the following digital repositories: PeerJ, PubMed Central SCIE and CLOCKSS.

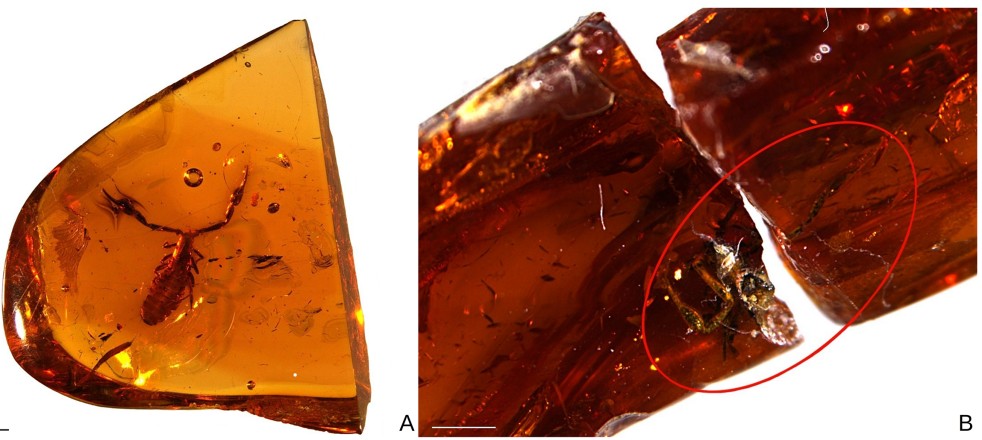

**Figure 4** *Baltamblyolpium* **gen. nov. inclusions in Baltic (A) and Bitterfeld (B) ambers.** (A) *Baltamblyolpium gizmotum* sp. nov., holotype ♀ (GPIH 05069 (ex. CGPC)). (B) *Baltamblyolpium grabenhorsti* sp. nov., holotype, adult (GZG.BST.30001 (ex HGPG (PS 12))). Scale bars = 2 mm.

## Systematics

Order Pseudoscorpiones de Geer, 1778
Suborder Iocheirata *Harvey, 1992*
Superfamily Garypinoidea *Daday, 1889*
Garypinidae *Daday, 1889*
Amblyolpiinae *Harvey, 2023*

**Baltamblyolpium gen. nov.**

Figures 1–12

ZooBank: urn:lsid:zoobank.org:act:82A24A38-BD7A-411D-8942-01E1AD8DD5EA

TYPE SPECIES: *Baltamblyolpium gizmotum* sp. nov.

ETYMOLOGY: The genus name comprises 'Balt-' for Baltic amber, the type locality of the type species *B. gizmotum*, and '*amblyolpium*' which refers to the close resemblance between the new genus, and *Amblyolpium* and *Neoamblyolpium*.

DIAGNOSIS: *Baltamblyolpium* differs from other genera of Garypinidae as follows: from *Neominniza* Beier, 1930, *Oreolpium Benedict & Malcolm, 1978*, *Protogarypinus* Beier, 1954, *Teratolpium Beier, 1959* and *Thaumatolpium Beier, 1931* by the divided arolia (Fig. 9); from *Aldabrinus Chamberlin, 1930*, *Caecogarypinus Dashdamirov, 2007*, *Galapagodinus Beier, 1978*, *Garypinidius Beier, 1955b*, *Garypinus Daday, 1889*, *Haplogarypinus Beier, 1959*, *Hemisolinus Beier, 1977*, *Indogarypinus Murthy & Ananthakrishnan, 1977*, *Nelsoninus Beier, 1967*, *Paraldabrinus Beier, 1966*, *Protogarypinus* Beier, 1954, *Pseudogarypinus Beier, 1931*, *Serianus Chamberlin, 1930* (junior synonym, *Paraserianus* Beier, 1939), *Solinellus Muchmore, 1979* and *Solinus Chamberlin, 1930* by the position of trichobothrium *it* which is situated in the distal half of the fixed finger; and from *Amblyolpium* and *Neoamblyolpium* by: the position of trichobothrium *st* which is situated closer to *sb* than to *t*; the position of trichobothrium *it* which is situated close to the tip of the fixed finger; the femora of legs I and II being only slightly shorter or equal to

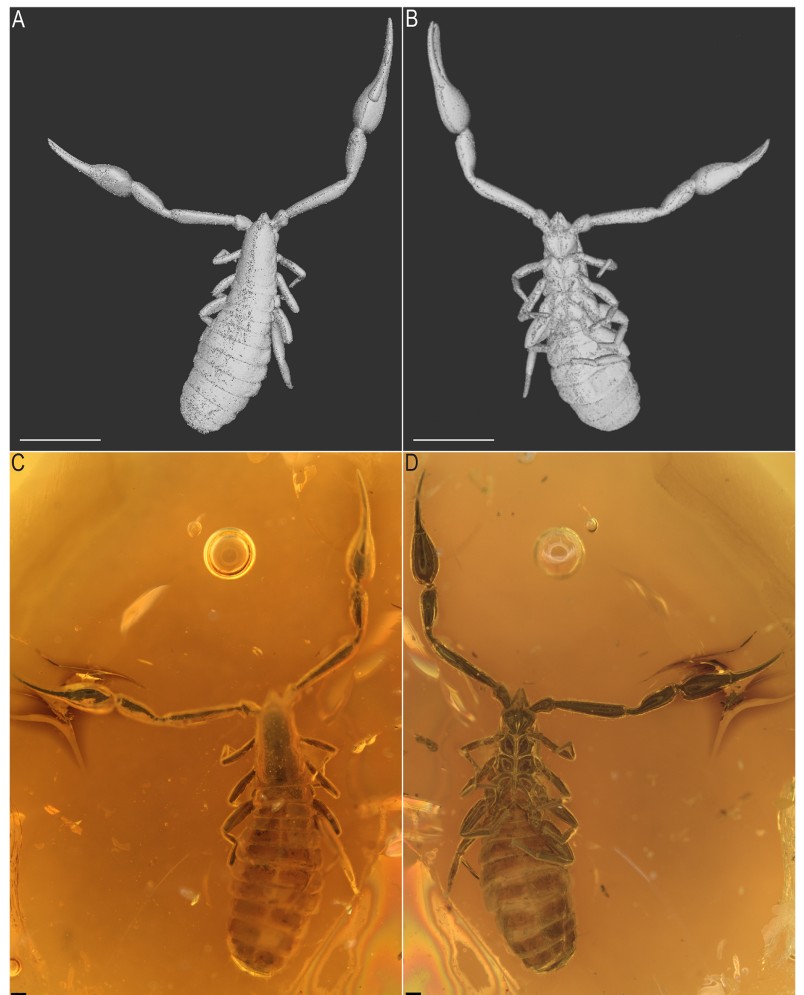

**Figure 5 *Baltamblyolpium gizmotum* sp. nov., holotype ♀ (GPIH 05069 [ex. CGPC]), dorsal (A, C) and ventral (B, D) habitus, as 3D-reconstruction models (A, B) and photographs (C, D).** Scale bars: A, B = 1 mm; C, D = 0.2 mm.

the patellae; and the presence of a pair of glandular setae on sternites VI–VIII, which are also situated anterior to the row of regular setae.

REMARKS: *Baltamblyolpium* does not match any of the Holocene genera of Garypinidae, although it most closely resembles *Amblyolpium* and *Neoamblyolpium* in the position of trichobothria *isb* and *ist* which are located in the medial region of the pedipalp finger, rather than sub-basally as in all other garypinids. As outlined in the Diagnosis, *Baltamblyolpium* differs from *Amblyolpium* and *Neoamblyolpium* in several ways. Trichobothrium *it* is situated in the distal half of the fixed chelal finger (Fig. 7), but is situated medially adjacent to *isb* and *ist* in *Amblyolpium* and *Neoamblyolpium*. Trichobothrium *st* is situated closer to *sb* than to *t* in *Baltamblyolpium*, but is closer to *t* in *Amblyolpium* and *Neoamblyolpium* (*e.g.*, *Heurtault, 1970*; *Lazzeroni, 1970*; *Harvey, 1988*; *van den Tooren, 2002*; *Nassirkhani, Shoushtari & Abadi, 2016*; *Nassirkhani & Doustaresharaf, 2019*). One of the most unusual differences lie in the shape of legs I and II in which the femora are about the same length as the patellae (Fig. 9), which corresponds

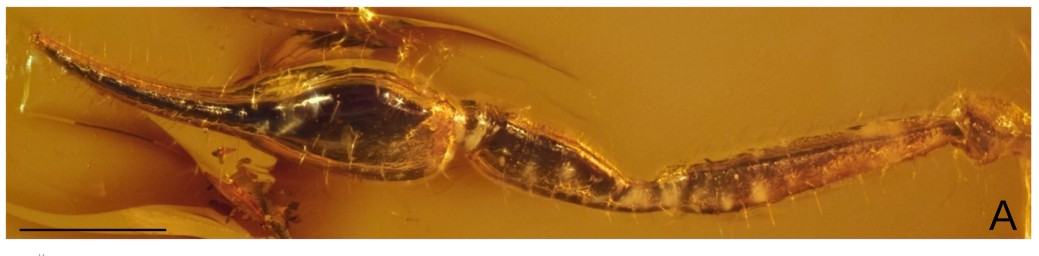

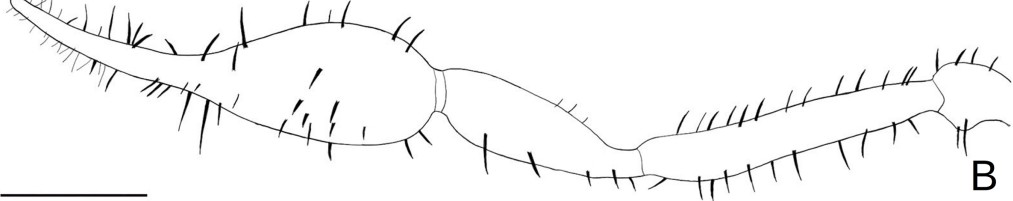

**Figure 6 *Baltamblyolpium gizmotum* sp. nov., holotype ♀ (GPIH 05069 [ex. CGPC]), pedipalp as photograph (A) and interpretative drawing (B).** Scale bars = 0.5 mm.

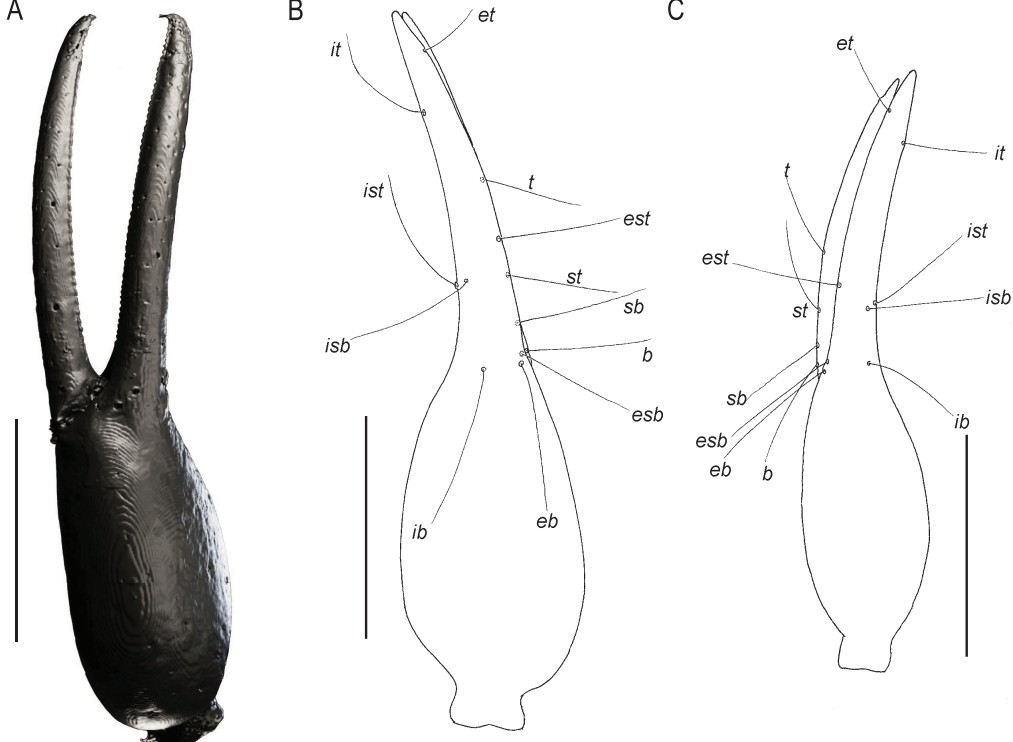

**Figure 7 Chelae of *Baltamblyolpium* gen. nov.** (A, B) *Baltamblyolpium gizmotum* sp. nov., holotype ♀ (GPIH 05069 (ex. CGPC)), left (A) and right (B) chelae, retrolateral (A) and dorsal (B) aspects. (C) *Baltamblyolpium grabenhorsti* sp. nov., holotype, adult (GZG.BST.30001 (ex HGPG (PS 12))), left chela, dorsal aspect. (A) 3D-reconstruction model. (B, C) Interpretative drawings drawn to scale. Scale bars = 0.5 mm.

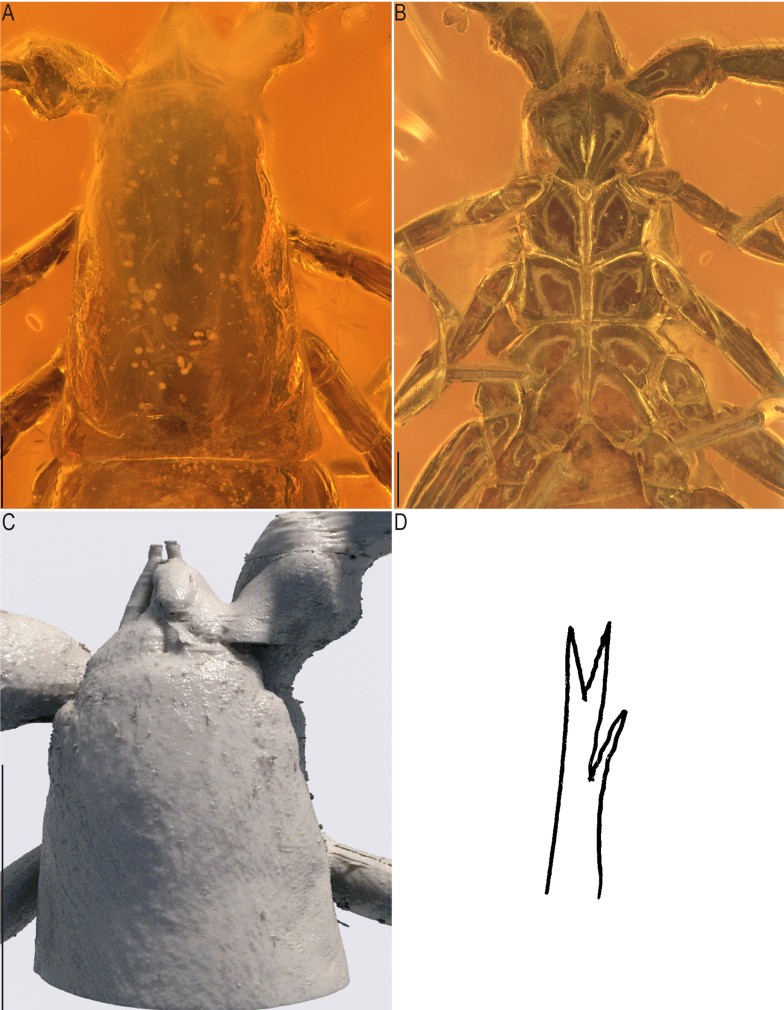

**Figure 8 *Baltamblyolpium gizmotum* sp. nov., holotype ♀ (GPIH 05069 (ex. CGPC)), carapace and chelicerae.** (A) Photograph of carapace and chelicerae; (B) photograph of sternites; (C) 3D-reconstruction model of carapace and chelicerae; (D) interpretive drawing of galea. Scale bars: A, B = 0.2 mm; C = 0.5 mm.

to the condition found in most garypinids with the exception of *Amblyolpium* and *Neoamblyolpium* where the femora are noticeably longer than the patellae (*e.g.*, *Heurtault, 1970*; *Lazzeroni, 1970*; *Harvey, 1988*; *van den Tooren, 2002*; *Nassirkhani, Shoushtari & Abadi, 2016*; *Nassirkhani & Doustaresharaf, 2019*). The final difference lies in the disposition of the pair of glandular setae on the abdominal sternites. In *Amblyolpium* and *Neoamblyolpium* they are only located on sternites VI and VII and situated in line with the regular setal row. In *Baltamblyolpium*, they are located on sternites VI, VII and VIII and situated noticeably anterior to the regular setal row very close to the median suture line (Fig. 10).

Included Taxa: *Baltamblyolpium gizmotum* sp. nov. and *Baltamblyolpium grabenhorsti* sp. nov.

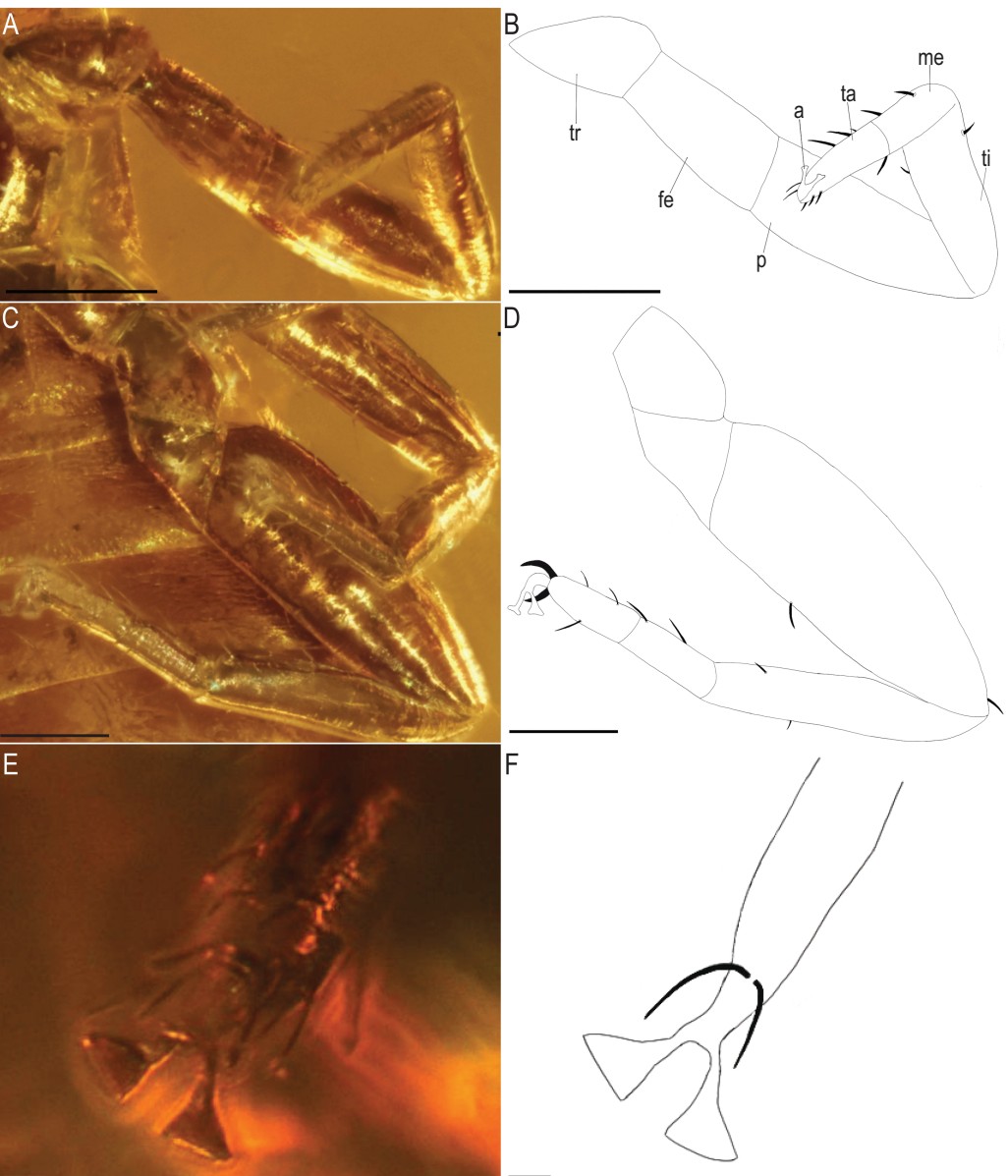

**Figure 9** *Baltamblyolpium gizmotum* sp. nov., holotype ♀ (GPIH 05069 (ex. CGPC)), leg I (A, B), leg IV (C, D), arolium (E, F) as photographs (A, C, E) and interpretive drawings (B, D, F). Scale bars: A, B, C, D = 0.2 mm; E, F = 0.02 mm. tr, trochanter; fe, femur; p, patella; ti, tibia; me, metatarsus; ta, tarsus; a, arolium.

*Baltamblyolpium gizmotum* sp. nov.

Figures 1–3, 4A, 5, 6, 7A, 7B, 8–10

ZooBank: urn:lsid:zoobank.org:act:FD329941-9416-4758-939B-A58C366A59A1

MorphoSource: ark:/87602/m4/490391; ark:/87602/m4/490395; ark:/87602/m4/490399

TYPE MATERIAL: Holotype ♀ (GPIH 05069 [ex. CGPC]), Baltic amber from Lithuania.

ETYMOLOGY: The specific epithet refers to the artist name, Gizmo, of the first author's father.

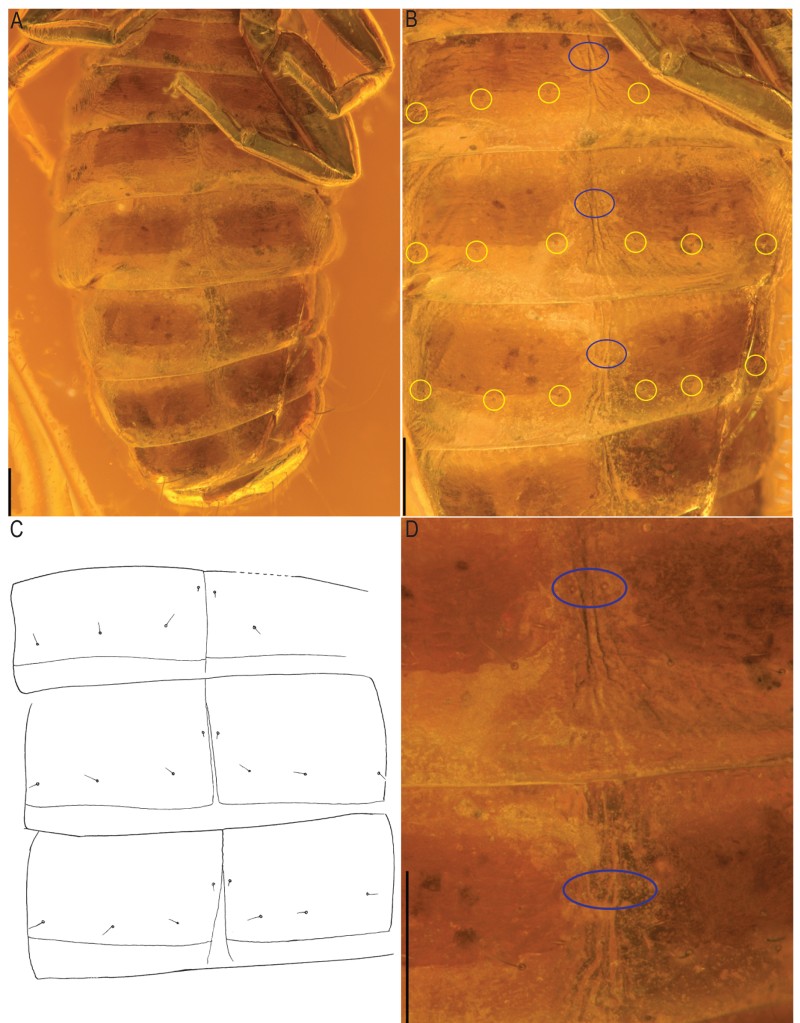

**Figure 10 *Baltamblyolpium gizmotum* sp. nov., holotype ♀ (GPIH 05069 (ex. CGPC)), sternites, IV–X (A), VI–VIII (B, C), VII, VIII (D) as photographs (A, B, D) and interpretive drawing (C).** Scale bars = 0.2 mm.                     

DIAGNOSIS: *B. gizmotum* differs from *B. grabenhorsti* sp. nov. by its larger size, *e.g.*, chela (including pedicel) 2.18 mm *vs.* 1.34 mm in *B. grabenhorsti*; and the position of trichobothrium *t* which is more distally placed in *B. gizmotum* compared with *B. grabenhorsti*.

DESCRIPTION: The following description is based on the holotype female, the only known specimen (Figs. 4A, 5, 6, 7A, 7B, 8–10). The specimen is adult, which is evident through the presence of four trichobothria on the movable chelal finger.

*Color* (in amber): Body reddish brown; pedipalps, carapace and legs darker than rest of body.

*Prosoma*: Carapace slender (1.75× longer than wide); four simple corneate eyes; carapace with 20 setae, arranged in 4: 4: 4: 4: 4 (Fig. 8) and without transverse furrow.

*Pedipalps*: Pedipalp (including trochanter + femur + patella + chela with pedicel length) length/width 4.10/0.24 (Fig. 6); chela long and slender (chela with pedicel length: width

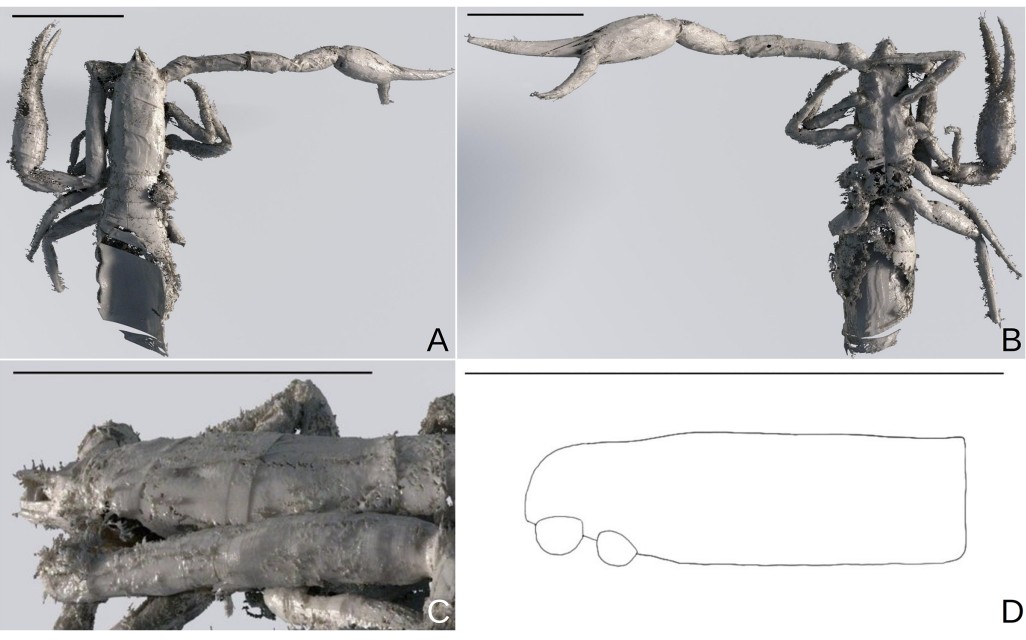

**Figure 11** *Baltamblyolpium grabenhorsti* **sp. nov., holotype, adult (GZG.BST.30001 (ex HGPG (PS 12))), habitus (A, B) and carapace (C, D).** (A, B) 3D-reconstruction models of dorsal (A) and ventral (B) habitus. (C, D) Lateral carapace as 3D-reconstruction model (C) and interpretive drawing (D). Scale bars = 1 mm.

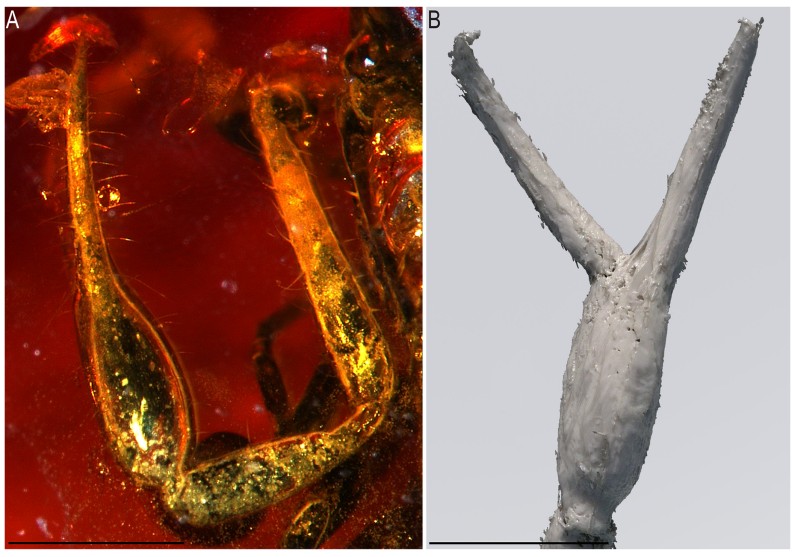

**Figure 12** *Baltamblyolpium grabenhorsti* **sp. nov., holotype, adult (GZG.BST.30001 (ex HGPG (PS 12))), pedipalp chela, dorsal (A) and retrolateral (B) aspects.** (A) Photograph. (B) 3D-reconstruction model. Scale bars = 0.5 mm.

ratio 4.13); 52 small rounded teeth on fixed finger, 38 small pointed teeth on movable finger, teeth juxtadentate; venom teeth present in both fingers, venom apparatus of movable finger not visible; fixed chelal finger with eight trichobothria, movable finger with four trichobothria (Figs. 7A, 7B); trichobothria: *eb* and *esb* subbasal; *est* medial and slightly

closer to *esb* than to *et*; *et* subdistal, close to tip of finger; *ib* distal to *eb* and *esb*; *it* distal to *est*; *st* closer to *sb* than to *t*; *sb* closer to *b* than to *st*; *t* situated slightly distal of the middle of the finger.

*Chelicerae*: More than twice as long as wide (chelicerae length: width ratio 2.25); galea long (0.04) with three rami, including one medial and two distal (Figs. 8A, 8C, 8D); hand with setae, but total number not observable; movable finger with one subdistal seta.

*Coxal area*: Pedipalpal coxa with 10 setae; coxae I–IV with setae arranged 10: 8: 4: 2.

*Opisthosoma*: 2.42× longer than carapace; tergites II–X and sternites IV–X with medial suture, others undivided; chaetotaxy of tergites I–XII: 4: 4: 4: 4: 4: 4: 4: 6: 4: 10: ?: 2; chaetotaxy of sternites II–XII: : ?: 6: 4: 6: 6: 6: 6: 8?: 2; pleural membrane without setae; sternites VI–VIII with a single median pair of glandular setae situated well in advance of the regular setae; genital area simple (Fig. 8B).

*Legs*: Femora of legs I and II about same length as patellae; long seta on metatarsus of leg II; arolium distinctly longer than claws and deeply divided (Fig. 9); tarsal setae undivided; claws simple and slender.

*Dimensions*: Body length (including chelicerae) 3.92; prosoma 1.09; opisthosoma 2.20/1.06; carapace 1.14/0.65; pedipalps: trochanter 0.28/0.24, femur 1.04/0.21, patella 0.78/0.25, chela (without pedicel) 1.61/0.40, chela (including pedicel) 1.65; leg I: trochanter 0.49/0.33, femur 0.54/0.27, patella 0.94/0.31, tibia 0.65/0.20, metatarsus 0.35/0.16, tarsus 0.39/0.14; leg IV: trochanter 0.53/0.41, femur 0.50/0.30, patella 1.82/0.58, tibia 1.42/0.24, metatarsus 0.47/0.19, tarsus 0.56/0.16; chelicera 0.18/0.08, galea 0.04 (length).

REMARKS: This specimen is well-preserved and most morphological details are visible although some details (*e.g.*, cheliceral setation) are obscured by emulsions and could not be modeled using synchrotron data.

### *Baltamblyolpium grabenhorsti* sp. nov.

Figures 1–3, 4B, 7C, 11, 12
ZooBank: urn:lsid:zoobank.org:act:3D604398-704B-40B1-9DE8-7205DEEB4BDF
MorphoSource: ark:/87602/m4/490384

TYPE MATERIAL: Holotype, adult but sex unknown (GZG.BST.30001 [ex HGPG (PS 12)]), Bitterfeld amber, region around Goitzsche, Saxony, Germany.

ETYMOLOGY: The specific epithet is a patronym honoring Heinrich Grabenhorst, who kindly provided the specimen.

DIAGNOSIS: *B. grabenhorsti* differs from *B. gizmotum* sp. nov. by its smaller size, *e.g.*, chela (including pedicel) 1.34 mm *vs*. 2.18 mm in *B. gizmotum*; and the position of trichobothrium *t* which is more mediallly placed in *B. grabenhorsti* compared with *B. gizmotum*.

DESCRIPTION: The following description is based on the holotype, the only known specimen (Figs. 4B, 7B, 11, 12). The specimen is adult due to the presence of four trichobothria on the movable chelal finger (Figs. 7B, 12) but the opisthosoma is incompletely preserved and the specimen cannot be sexed.

*Color* (in amber): Dark brown to black, moderately oxidized.

*Prosoma*: Carapace slender (1.84× longer than wide); four corneate eyes; carapace setation not visible.

*Pedipalps*: Pedipalp (including trochanter + femur + patella + chela with pedicel length) length/width 2.76/0.19 (Figs. 7B, 12); chela long and slender (chela with pedicel length: width ratio 5.36); all chelal teeth seemingly juxtadentate, venom teeth present in both fingers; trichobothria: *eb* and *esb* subbasal; *est* not visible; *et* subdistal, close to tip of finger; *ib* situated basally at same level as *eb* and *esb*; *it* subdistal; *st* slightly closer to *sb* than to *t*; *sb* closer to *b* than to *st*; *t* situated slightly basal of the middle of the finger.

*Chelicerae*: Nearly twice as long as wide (chelicerae length: width ratio 1.83), galea (0.04 mm) with three rami, including one medial and two distal; two long setae and one short seta visible on right cheliceral hand, in addition one seta discernable on movable finger of left chelicera; serrula exterior and interior with at least 11 long blades.

*Opisthosoma*: Approximately 1.84× longer than carapace but imperfectly preserved; tergites with medial suture.

*Legs*: Femora of legs I and II about same length as patellae (Fig. 11); arolium distinctly longer than claws and deeply divided; claws simple and slender.

*Dimensions*: Body length (including chelicerae) ca. 2.74; carapace 0.92/0.45; opisthosoma ca. 1.69/0.63; pedipalps: trochanter 0.25/0.17, femur 0.80/0.13, patella 0.37/0.19, chela (without pedicel) 1.26/0.24, chela (including pedicel) 1.34/0.25, finger length 0.71; chelicera 0.11/0.06, galea length 0.04; leg I: trochanter 0.10/0.11, femur 0.21/0.19, patella 0.28/0.12, tibia 0.29/0.07, metatarsus 0.18/0.06, tarsus 0.20/0.05; leg IV: trochanter 0.13/0.15, femur 0.17/0.14, patella 0.62/0.17, tibia 0.41/0.09.

REMARKS: Due to imperfect preservation (the amber is cracked into two pieces and the opisthosoma is only partly preserved in the darkened amber) not all morphological characters, which may be important for generic identification, could be observed. However, there are several notable similarities with *B. gizmotum*, such as the femora of legs I and II being about the same length as the patellae, the shape of the pedipalps and trichobothrium *st* being situated midway between *sb* and *t*, and we confidently assign this species to *Baltamblyolpium*.

## DISCUSSION

### Systematics and morphological stasis of Garypinidae

The pseudoscorpion family Garypinidae was first described by *Daday (1889)* as a subfamily of Cheliferidae based on a single species, *Garypinus dimidiatus* (L. Koch, 1873). *Chamberlin (1930)* transferred Garypininae to Olpiidae *Chamberlin, 1930*, where it remained until it was treated as a distinct family (*Judson, 1992*, *1993*, *2005*). Since its initial description, Garypinidae has undergone multiple taxonomic changes, including the addition of new genera, synonymizations and genus transfers. The genera *Amblyolpium*, *Aldabrinus*, *Serianus*, *Solinus*, *Pseudogarypinus*, *Garypinidius*, *Haplogarypinus*, *Paralabrinus*, *Nelsoninus*, *Hemisolinus*, *Galapagodinus*, *Neoamblyolpium*, *Indogarypinus*, *Solinellus*, *Caecogarypinus*, were added by *Simon (1898)*, *Chamberlin (1930)*, *Beier (1931*, *1955b*, *1959*, *1966*, *1967*, *1977*, *1978)*, *Hoff (1956)*, *Murthy & Ananthakrishnan (1977)*, *Muchmore (1979)* and *Dashdamirov (2007)*, respectively. *Mahnert (1988)* synonymized

*Paraserianus* Beier, 1939 with *Serianus*. *Harvey & Šťáhlavský (2010)* transferred *Neominniza*, *Oreolpium*, *Protogarypinus*, *Teratolpium* and *Thaumatolpium* from Olpiidae to Garypinidae and re-diagnosed the family. *Harvey (2023)* proposed a new subfamily classification for Garypinidae. Although all garypinid genera are extant, two extinct species are known. *Cockerell (1920)* described *Gayrpus burmiticum* from Burmese amber, which *Judson (1997)* later transferred to *Amblyolpium*. Its status remains an enigma, however, given the poor state of the original description. A second extinct species, *Garypinus electri* from Baltic amber, was described by *Beier (1937)*. Although the type is likely lost (*Penney, 2003*), *Judson (2005)* discovered a second specimen of *G. electri* holding the leg of a tipuloid fly in Baltic amber, but noted that his specimen exhibited a long, submobile patella, whereas in *Beier (1937)*'s original description *G. electri* is described as having a short and mobile patella. *Judson (2005)* assumed this to be an error in the original description considering all *Garypinus* species exhibit the former condition. Presently, Garypinidae includes 81 extant and two extinct species in 21 extant genera.

Phylogenetic analyses of pseudoscorpion families based on morphological (*Harvey, 1992*) and multi-locus sequence data (*Murienne, Harvey & Giribet, 2008*) place Garypinidae in Iocheirata, a monophyletic suborder comprising all pseudoscorpions with venom glands. However, based on multi-locus and transcriptomic analyses, Garypinidae is rendered paraphyletic by the Holarctic, monogeneric family Larcidae *Chamberlin, 1930* and both families are the only members of the monophyletic superfamily Garypinoidea (*Murienne, Harvey & Giribet, 2008*; *Benavides et al., 2019*; *Harvey, 2023*). A dated phylogenomic analysis of pseudoscorpion transcriptomes calibrated using fossils recovered a Late Jurassic origin (ca. 150 Ma) for this superfamily (*Benavides et al., 2019*). A Mesozoic origin for the superfamily is also plausible considering that *Amblyolpium* has been recorded in Cretaceous Burmese amber (Fig. 2). The presence of garypinids in Burmese amber confirms that all diagnostic features of the family were established by the Upper Cretaceous (Cenomanian: 99 million years ago). *Baltamblyolpium* gen. nov. from European succinite ambers represents the third fossil record and first extinct genus of Garypinidae.

The description of *Baltamblyolpium* reinforces the hypothesis of morphological stasis within Garypinidae since the Mesozoic because characters dividing this extinct genus from other genera are minor. *Baltamblyolpium* gen. nov. can be placed confidently in Garypinidae due to the following combination of characters: divided arolia; trichobothria *isb* situated on the prolateral margin of the chelal fingers; glandular setae present on the medial sternites; and four eyes on the prosoma. It does not resemble *Garypinus*, the only other garypinid genus known from Baltic amber, but rather is most similar to the widespread genus *Amblyolpium*, which is found in southern Europe, the Middle East, South and Southeast Asia, Melanesia, Africa, the Caribbean and South America, and to *Neoamblyolpium* from western North America. All three genera share the medial position of trichobothria *isb* and *ist* on the chelal finger, rather than a sub-basal position as in all other garypinids (Fig. 7). Although we are unable to suggest a sister-relationship between *Baltamblyolpium* and any genus, some conclusions are possible: (i) *Baltamblyolpium* has strong morphological affinities to a genus (*Amblyolpium*) whose distribution in Europe is

confined to the Mediterranean Basin; and (ii) Garypinidae has remained relatively unchanged morphologically between the Cretaceous and Eocene, and between the Eocene and today. Phylogenetic analysis of morphological and molecular data, however, is required to help resolve relationships among extant garypinids and determine the placement of fossil taxa.

## Garypinidae experiences extinction and range contraction in Europe during Cenozoic climate change

The Mediterranean Basin is a biodiversity hotspot with great diversity and a high proportion of endemic plant and animal species (*Médail & Diadema, 2009*; *Krause et al., 2022*). This high diversity and endemism are best explained by the climatic and geological history of this region. During the Eocene–early Pliocene, Europe experienced a warm-temperate to subtropical climate (*Sadowski et al., 2017*). However, beginning in the late Pliocene, significant cooling occurred (*Kvaček, Teodoridis & Denk, 2020*) and by the Last Glacial Maximum (18,000 years ago), large parts of Europe were covered with ice sheets, tundra and cold steppe (*Bilton et al., 1998*; *Brito, 2005*). According to the 'Mediterranean glacial refugium' hypothesis, the Mediterranean Basin served as a refugium for many Central European taxa during Plio-Pleistocene glaciation. Across vertebrates (*Sommer & Zachos, 2009*; *Perktaş, Gür & Ada, 2015*) and plants (*Gentili et al., 2015*; *Di Pasquale et al., 2020*), molecular phylogenetic analyses and fossil data indicate that many taxa were widespread across Central Europe prior to the Last Glacial Maximum, became restricted to southern Europe during glaciation, and experienced range expansion once climatic conditions became more stable.

The discovery of the two *Baltamblyolpium* species and occurrence of *G. electri* in Baltic amber clearly demonstrate that Garypinidae was present in northern Europe during the Eocene (Fig. 1). The absence of Garypinidae in Central Europe and the presence of several genera (*Garypinus*, *Amblyolpium*, *Serianus*, *Solinus*) in southern Europe and/or northern Africa today (Fig. 3) support the 'Mediterranean glacial refugium' hypothesis for this family and perhaps many northern European garypinids (*e.g.*, *Garypinus*) retreated to the Mediterranean Basin during Quaternary glaciation, whereas some (*e.g.*, *Baltamblyolpium*) went extinct. However, garypinids, unlike many other Central European lineages seeking climatic refuge in the Mediterranean, did not expand back into Central Europe following the end of Pleistocene glaciation. This pattern of range contraction without expansion may have occurred across their distribution in the Holarctic as garypinids in the Northern Hemisphere are restricted to low mid-latitudes (<44°N). The biogeographical scenario for Garypinidae is supported through comparison with several other pseudoscorpion lineages. For example, three species of Geogarypidae are found in Baltic and Rovno ambers, yet the family is presently confined to southern Europe. Within Syarinidae *Chamberlin, 1930*, although no members have been recorded in European ambers, *Pseudoblothrus Beier, 1931* exhibits a relictual distribution as it is only known from caves in the Azores Archipelago (Portugal), Crimean Peninsula (Ukraine), France, Italy, and Switzerland (*Turbanov & Kolesnikov, 2020*; *WPC, 2023*). In contrast, Pseudotyrannochthoniidae occurs in both Baltic and Bitterfeld ambers but is absent in Europe today and its current distribution in

the Holarctic includes forest or mountain refugia in Central and East Asia, and the western U.S.A. (*Schwarze et al., 2021*; *Harvey & Harms, 2022*; *You et al., 2022*). *Pseudogarypus* Ellingsen, 1909 (Pseudogarypidae) is diverse in European succinite ambers with five species (*Harms & Dunlop, 2017*), but today known only from forest refugia and caves in the western and eastern U.S.A., respectively (*Harvey & Šťáhlavský, 2010*). Based on these examples, it appears that since the Paleogene, some pseudoscorpion lineages (Pseudotyrannochthoniidae, *Pseudogarypus* and *Baltamblyolpium*) went extinct in Europe, whereas others survived in refugia in the Mediterranean Basin (*Garypinus*, Geogarypidae) or in European caves (*Pseudoblothrus*). Some invertebrate lineages found in the European succinite ambers exhibit more extreme extinction patterns with modern taxa completely absent from northern latitudes. The pseudoscorpion family Feaellidae includes a single species in Baltic amber and another species from Triassic deposits in Ukraine (*Kolesnikov et al., 2022*) but today is only known from the Southern Hemisphere (Africa, Madagascar, Australia, India, the Seychelles, Sri Lanka, Maldives, Southeast Asia and Brazil; *Novák, Lorenz & Harms, 2020*; *Lorenz et al., 2022*). A similar pattern is observed in the spider family Archaeidae *Koch & Berendt, 1854*, which is found today in southern Africa, Madagascar and Australia (*Wood et al., 2013*), and the insect family Mantophasmatidae Klass, 2002 that occurs today in sub-Saharan Africa (*Klass et al., 2002*). Comparative analysis of dated phylogenies that include both fossil and extant taxa is required to elucidate the underlying causes of these various biogeographical patterns.

### *Baltamblyolpium*—a dry-adapted arachnid in the "Baltic amber forest"?

The "Baltic amber forest" of the Eocene was a dynamic ecosystem with a plethora of habitat types ranging from warm and humid, temperate conifer-angiosperm forests to coastal swamps, river systems and open habitats (*Sadowski et al., 2017*, *2019*). Pseudoscorpions preserved in succinite ambers reflect this habitat diversity and include both corticolous (bark-dwelling, *e.g.*, Atemnidae, Cheliferidae, Chernetidae) and humicolous (leaf litter, *e.g.*, Chthoniidae, Neobisiidae, Feaellidae, Geogarypidae, Pseudotyrannochthoniidae, Pseudogarypidae) lineages (*Benedict & Malcolm, 1970*, *1978*; *Harms & Harvey, 2013*; *Harms & Dunlop, 2017*; *Schwarze et al., 2021*). Phoresy, the ability to hitchhike on other organisms for transport, has also been documented in pseudoscorpions preserved in Baltic amber (*e.g.*, Geogarypinidae, Lechytiidae, Chernetidae, Cheliferidae, Withiidae), and in fact, the second fossil of *G. electri* in Baltic amber was attached to the leg of a fly, implying phoresy in Garypinidae dates at least to the Eocene (*Judson, 2005*). However, making further inferences about the ecology of *Baltamblyolpium* is challenging given that extant garypinids include both corticolous and humicolous species. *Amblyolpium* and *Neoamblyolpium*, the genera most closely resembling *Baltamblyolpium*, also have diverse habitat and microhabitat preferences. *Amblyolpium* is a widespread, speciose genus that includes both corticolous and humicolous species (*Nassirkhani, Shoushtari & Abadi, 2016*), and more than 60% of species are found in dry biomes (*e.g.*, deserts, open savannas, Mediterranean), whereas others are restricted to tropical and temperate forests. In contrast, *Neoamblyolpium* can be

described as a humicolous genus, with species recorded from the leaf litter of juniper, pinyon, yellow-pine, mountain mahogany trees in dry, high altitude (1,820–2,530 m) habitats in western North America (Colorado, New Mexico and California only) (*Hoff, 1956*). Given the abundance of *Amblyolpium* and *Neoamblyolpium* species in arid habitats, it remains a possibility that *Baltamblyolpium* represents another rare example of a more dry-adapted fauna present in Mediterranean or even drier regions in the "Baltic amber forest" ecosystem, perhaps similar to the Baltic amber pseudoscorpion, *Feaella groehni Henderickx & Boone, 2014* (Feaellidae), whose extant relatives are only known today from dry habitats at southern latitudes on former Gondwanan landmasses (*Henderickx & Boone, 2014*; *Harms & Dunlop, 2017*). The Baltic amber solifuges, *Palaeoblossia groehni Dunlop, Wunderlich & Poinar, 2004* and *Eognosippus fahrenheitiana Dunlop, Erdek & Bartel, 2023* in the family Daesiidae Kraepelin, 1899 (*Dunlop, Wunderlich & Poinar, 2004*; *Dunlop, Marusik & Vlaskin, 2019*; *Dunlop, Erdek & Bartel, 2023*), are other potentially dry-adapted taxa as Daesiidae is widely distributed in arid regions of Africa, the Middle East and southern Europe today. If *Baltamblyolpium* was dry-adapted, it might therefore provide additional evidence for the presence of open, xeric habitats in the "Baltic amber forest" that co-existed among closed, mesic forests in this complex Eocene ecosystem. Alternatively, it is equally possible that this genus was forest-dwelling.

## CONCLUSIONS

The present contribution describes a new genus of pseudoscorpions, *Baltamblyolpium* gen. nov., and two new fossil species of the pseudoscorpion family Garypinidae from the European succinite ambers of the Eocene: *Baltamblyolpium gizmotum* sp. nov. from Baltic amber and *Baltamblyolpium grabenhorsti* sp. nov. from Bitterfeld amber. The discovery represents the first extinct genus of Garypinidae and second record of Garypinidae in European Succinite ambers. *Baltamblyolpium* most closely resembles *Amblyolpium*, which is widespread but includes taxa restricted to Mediterranean refugia in Europe, and *Neoamblyolpium* from western North America. Because diagnostic characters separating this new fossil genus from other genera are minor, the discovery confirms morphological stasis within Garypinidae since the Mesozoic—a pattern that has been observed in many other pseudoscorpion lineages. The presence of Eocene fossils in northern Europe suggests that Garypinidae had a wider distribution across Europe, however, extinction and range contraction into Mediterranean refugia during the late Cenozoic (rather than dispersal) resulted in their present-day southerly European distribution as many other lineages that once thrived in the European "Baltic amber forest" of the Eocene.

## ACKNOWLEDGEMENTS

We thank the following colleagues: Carsten Gröhn (Glinde, Germany) and Heinrich Grabenhorst (Wienhausen, Germany) for donating specimens from their collections; Alexander Gehler from the Geosciences Museum, University of Göttingen for registration numbers; Carsten Gröhn for assistance with specimen preparation; Jithin Johnson and Dora Hlebec for assistance with morphological characters; and Jithin Johnson, Valentin Ehrenthal and four reviewers for invaluable comments on an earlier draft of this

manuscript. We thank colleagues at PETRAIII at the German Electron Synchrotron (Deutsches Elektronen-Synrchrotron) for assistance with scanning.

### Funding

Funding for this research was provided by the German Science Foundation award HA 8785/5 and KO 3944/10 to Danilo Harms and Ulrich Kotthoff. Synchrotron scans were generated with support of Elektronen-Synchrotron of the Helmholtz Association within the framework of a PETRA III regular proposal (BAG-20210019) for beamtime to scan amber fossils. The funders had no role in study design, data collection and analysis, decision to publish, or preparation of the manuscript.

### Grant Disclosures

The following grant information was disclosed by the authors:
German Science Foundation: HA 8785/5 and KO 3944/10.
Elektronen-Synchrotron of the Helmholtz Association: BAG-20210019.

### Competing Interests

The authors declare that they have no competing interests.

### Author Contributions

- Nova Stanczak performed the experiments, analyzed the data, prepared figures and/or tables, authored or reviewed drafts of the article, and approved the final draft.
- Mark S. Harvey performed the experiments, analyzed the data, prepared figures and/or tables, authored or reviewed drafts of the article, and approved the final draft.
- Danilo Harms conceived and designed the experiments, authored or reviewed drafts of the article, and approved the final draft.
- Jörg U. Hammel performed the experiments, prepared figures and/or tables, and approved the final draft.
- Ulrich Kotthoff conceived and designed the experiments, prepared figures and/or tables, and approved the final draft.
- Stephanie F. Loria conceived and designed the experiments, performed the experiments, analyzed the data, prepared figures and/or tables, authored or reviewed drafts of the article, and approved the final draft.

### Data Availability

The scans are available at MorphoSource: Project ID: 000490376; 10.17602/M2/M490391; 10.17602/M2/M490395; 10.17602/M2/M490399; 10.17602/M2/M490384.

### New Species Registration

The following information was supplied regarding the registration of a newly described species:

Publication LSID: urn:lsid:zoobank.org:pub:8EC926A6-3F0B-46A9-BD5A-CF79CB4326B8

*Baltamblyolpium*: urn:lsid:zoobank.org:act:82A24A38-BD7A-411D-8942-01E1AD8DD5EA

*Baltamblyolpium gizmotum*: urn:lsid:zoobank.org:act:FD329941-9416-4758-939B-A58C366A59A1

*Baltamblyolpium grabenhorsti*: urn:lsid:zoobank.org:act:3D604398-704B-40B1-9DE8-7205DEEB4BDF

## Supplemental Information

Supplemental information for this article can be found online at http://dx.doi.org/10.7717/peerj.15989#supplemental-information.

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
