# Peer review of "A new pseudoscorpion genus (Garypinoidea: Garypinidae) from the Eocene supports extinction and range contraction in the European paleobiota"

_PeerJ, doi:10.7717/peerj.15989_

## Round 0.1 · original submission · Minor Revisions

Dear Dr. Loria,

After this new review round and after one of the reviewers of the manuscript became an author, your manuscript received quite positive feedback from the new batch of reviewers. In summary, your manuscript has received 1 "accept" and 3 "minor review" decisions.

Still, please note that there are some improvements that need to be made before the manuscript is accepted for publication. Specifically, reviewer #3 raised some concerns regarding Baltamblyolpium grabenhorsti sp. nov. as a new species.

Considering all suggestions that have been made, I believe your manuscript deserves a minor review before it is accepted for publication in PeerJ.

Sincerely,
Daniel Silva

·

Basic reporting

This revised submission on two new pseudoscorpions in Eocene Baltic and Bitterfeld amber respectively makes a valuable contribution to our understanding of the fossil record of this group. The authors describe a new genus and two new species, which represent the first extinct genus discovered in the living family Garypinidae. The descriptions are detailed and of high quality and the research team involves several experienced experts on pseudoscorpion systematics and on amber in general. The authors also place their results in a wider context by discussing the biogeographical implications of their finds and reviewing the distribution of fossil and living pseudoscorpions in general.

The authors have addressed the main issue with the previous version of this manuscript - namely that the family identification appeared to be incorrect - and have amended the text accordingly throughout, while making the reviewer who pointed out this error (Harvey) an author. I think this is acceptable and certainly improves the overall scientific quality and accuracy of the paper.

Minor issues raised in the previous round of reviews appear to have been addressed appropriately.

Experimental design

The manuscript does not include experiments per se, but the imaging methods used (light microspopy and synchrotron reconstructions) are excellent and have resulted in a series of very high quality figures which document the new material in some detail.

The type specimens are deposited in recognised public institutions (Hamburg and Göttingen) which would allow them to be reinvestigated in future.

Validity of the findings

The manuscript makes a valid contribution to our understanding of the historical biogeography of pseudoscorpions. The authors proposal that the family Garypinidae has undergone range contraction after the Eocene appears to be well-justified and the distribution map of the living representatives is very helpful for underlining this point.

Comparative data from other pseudoscorpion families (and from arachnids in general) is discussed, and I think this study would also be of wider interest to workers on other groups found in European ambers. It may even be of use to readers more generally interested in faunal changes in response to the ice ages.

Additional comments

On line 229 there is a non-italicised "0" in Hertault, 1970, but this could be corrected at proof stage.

Otherwise I have no further comments beyond congratuating the authors on a nice piece of work which I think is now acceptable for final editing and publication.

Reviewer 2 ·

Basic reporting

Nothing negative to report, apart from the use of the noun "discovery" in the title and the concordance of gender (masculine vs neuter) in the first species described. Other minor comments/corrections are in the text.

Experimental design

No comment

Validity of the findings

No comment

Additional comments

Nothing negative to report, apart from the use of the noun "discovery" in the title and the concordance of gender (masculine vs neuter) in the first species described. Other minor comments/corrections are in the text.

Annotated reviews are not available for download in order to protect the identity of reviewers who chose to remain anonymous.

Reviewer 3 ·

Basic reporting

The manuscript is very well written. All parts are clear. I can only recommend breaking up some very long sentences in the introduction. This could contribute to an even better understanding of the text.
The manuscript includes all important references to the issue. It includes not only purely taxonomic articles, but also works allowing more general interpretations. All the literature are appropriately referenced.
The structure of the article reflects the fact that it contains descriptions of two new species. Therefore, detailed image documentation is essential. I would recommend adding insets with details, especially in the 3D-reconstruction images. The described structures are not visible very frequently. The colors of the symbols for the different genera in Figure 3 is not always well distinguishable. It can be recommended to include also different shapes. For Figure 7, I recommend unifying the magnification for a better comparison and reducing the size of the trichobothria symbols, which now seem rather distracting.

Experimental design

The manuscript represents a stand-alone study enabling a better understanding of diversity in the historical context of Earth. For this reason, it meets the requirements for publication in PeerJ.
The authors describe a new genus of fossil pseudoscorpions and place it in the framework of the current taxonomy of the studied group and also in the more general framework of the development of diversity in Europe during the Cenozoic. These fossils allow us to fill in the gaps of our knowledge about this topic.
The authors use available methodological approaches for the study of fossils including 3-D reconstruction using micro-computed tomography with Electron Synchrotron. Surprisingly, this method does not yield any more detailed information (at least it is not evident from the presented images). However, it is annoying if it remains included in the study. It would be ideal to add more detailed pictures. Ideally like interpretive drawings, where the characteristics important for the taxonomy of pseudoscorpions are better visible.

Validity of the findings

Description of the species Baltamblyolpium grabenhorsti sp. nov. seems to me disputable. It is based on a poorly preserved specimen, where it is probably not possible to specify some details very well (e.g. sex). This species lacks good pictorial documentation comparable to B. gizmotus sp. nov. The difference between these two species is based on the different size. However, this may be due to a difference between the sexes, as is the case in the recent species Serianus patagonicus (See Beier 1964b in World Arachnida Catalog) from the same family. Or it may also reflect the geographical and age differences of the two finds. The second mentioned diagnostic character "position of trichobothrium t" is also debatable. In B. gizmotus sp. nov. this position is obviously variable on the left and right pedipalp (Fig. 7b,c). I think it is more appropriate to interpret the damaged specimen as Baltamblyolpium sp. and take into account in the discussion that it may be a second species or a male of B. gizmotus sp. It is also appropriate to supplement this specimen with better visual documentation.
Furthermore, in the discussion, I cannot agree with the authors with their interpretation of the genus Baltamblyolpium as a "dry-adapted arachnid". There is no real indication that this is an example of a more dry-adapted fauna. The authors themselves point out that some genera of the Garypinidae family can live in both dry and wetter places and that there are different microhabitat preferences within scorpions (even within the mentioned corticolous and humicolous lineages). I think this part of the discussion should be reworked to take into account the fact that at the moment we cannot really specify the microhabitat preferences of the genus Baltamblyolpium.

Additional comments

I fully support the acceptance of the submitted manuscript after review of some parts. I recommend carefull consideration Baltamblyolpium grabenhorsti as a new species. At the moment, the mentioned differences do not seem to me to be sufficient and are poorly documented by figures. The interpretation of dry-adaptation does not seem relevant to me either.

Reviewer 4 ·

Basic reporting

Dear Editor,

The manuscript under review presents the description of a new genus of pseudoscorpions (Baltamblyolpium gen. nov) (Garinipidae), based on the description of two new species, Baltamblyolpium gizmotus sp. nov. nov. from Baltic amber and Baltamblyolpium grabenhorsti sp. nov. nov. from Bitterfeld amber. The new genus resembles Neoamblyolpium from western North America and also the genus Ambplyolpium, which originates mainly from the Mediterranean refuge in Europe. The discovery of these two species indicates that the family Garinipidae has undergone little change since the Mesozoic. Moreover, according to the authors, the presence of this fossil in northern Europe indicates that this family of pseudoscorpions should have had a wider geographic distribution in the Eocene. The climatic changes that have occurred since that time would have caused processes of extinction or retreat.

In my opinion, the manuscript is very well organized and presented. The description is adequate for both species. The results are surprising and exciting, and the analysis of extinction processes is very well researched and presented. The images of the fossils are of good quality, as are the illustrations and the digital reconstruction of the specimens. Therefore, I believe that the manuscript is close to acceptance for publication. However, I have minor comments on the text.

I really appreciate the opportunity.

Subjects. The term Biodiversity is duplicated.
Keywords: The terms biogeography, paleontology, systematics, Taxonomy may seem too broad or vague, or overlap with subjects.
P7. Line 87. Please, change (Sadowski et al., 2017; Sadowski et al., 2019) to (Sadowski et al., 2017; 2019).
P7. Lines 93-98. The sentence “Recent phylogenomic analyses” seems hard to follow. Perhaps it can be divided into two or three sentences.
P8. Lines 138-145. The two fossils were obtained from the Carsten Gröhn Private Collection (CGPC) and Heinrich Grabenhorst Private Collection (HGPC). Is there information about the provenance of the samples? When were the fossils obtained?
P10-11. This is a small detail, but the numbers of the figures, (e.g. Figs. 5-10) are cited without ascending order.
Figure 3. The crosses are difficult to locate in the image. Depending on the final resolution of the image in the digital version of the article, they may be difficult to see.

Experimental design

The experimental design seems to be adequate.

Validity of the findings

The description of the two new species proves to be adequate. The findings and conclusions about extinction events experienced by these groups as a result of environmental and climatic transformations are well supported.

---

## Round 0.2 · accepted · Accept

Dear Dr. Loria,

I am pleased to inform you that your manuscript has been formally accepted for publication in PeerJ. Congratulations on your hard work! Please make sure the project in Morphosource with raw 3D data listed in the manuscript becomes available before publication.

Sincerely,
Daniel Silva

Reviewer 3 ·

Basic reporting

The manuscript is even better understood and well prepared after revision and all important references are included. The Figure 3 is much clearer now. The 3D reconstruction seems to me to be still slightly disputable, and one can only hope that the promised files in MorphoSource really bring visible details, as the authors promised in response to the review, but are not visible in the images in the manuscript. Unfortunately, the mentioned project in MorphoSource is not yet accessible and it is difficult for the reviewers to assess whether the promised details are really in sufficient resolution. I have no choice but to believe that it is so.
I still have some doubts about figure 7. Correcting the legend has led me to believe that these are probably really two different species. However, the length of the chela on Figure 7A and 7B appears to be the same, but the lengths of the scale bars are apparently very different. Are they really 0.5 mm in both cases? Please, control it.

Experimental design

The experimental design is completely adequate for the study of fossil material.

Validity of the findings

After the correction of the Figure 7 I agree that Baltamblyolpium grabenhorsti sp. nov. may be a new species.

Additional comments

I fully support the acceptance of the submitted manuscript.